# Enhanced Antibacterial Activity of a Cationic Macromolecule by Its Complexation with a Weakly Active Pyrazole Derivative

**DOI:** 10.3390/biomedicines10071607

**Published:** 2022-07-06

**Authors:** Anna Maria Schito, Debora Caviglia, Chiara Brullo, Alessia Zorzoli, Danilo Marimpietri, Silvana Alfei

**Affiliations:** 1Department of Surgical Sciences and Integrated Diagnostics (DISC), University of Genoa, Viale Benedetto XV, 6, 16132 Genoa, Italy; amschito@unige.it; 2Department of Pharmacy (DIFAR), University of Genoa, Viale Cembrano, 16148 Genoa, Italy; chiara.brullo@unige.it; 3Cell Factory, IRCCS Istituto Giannina Gaslini, Via Gerolamo Gaslini 5, 16147 Genoa, Italy; alessiazorzoli@gaslini.org (A.Z.); danilomarimpietri@gaslini.org (D.M.)

**Keywords:** pyrazole compounds, CB1H-loaded copolymer NPs, Gram-positive and Gram-negative MDR isolates, MICs and MBCs, time-kill experiments, cytotoxicity on human cells, selectivity index

## Abstract

Molecules containing the pyrazole nucleus are widely reported as promising candidates to develop new antimicrobial compounds against multidrug-resistant (MDR) bacteria, where available antibiotics may fail. Recently, aiming at improving the too-high minimum inhibitory concentrations (MICs) of a pyrazole hydrochloride salt (CB1H), CB1H-loaded nanoparticles (CB1H-P7 NPs) were developed using a potent cationic bactericidal macromolecule (P7) as polymer matrix. Here, CB1H-P7 NPs have been successfully tested on several clinical isolates of Gram-positive and Gram-negative species, including relevant MDR strains. CB1H-P7 NPs displayed very low MICs (0.6–4.8 µM), often two-fold lower than those of P7, on 34 out of 36 isolates tested. Upon complexation, the antibacterial effects of pristine CB1H were improved by 2–16.4-fold, and, unexpectedly, also the already potent antibacterial effects of P7 were 2–8 times improved against most of bacteria tested when complexed with CB1H. Time-killing experiments performed on selected species established that CB1H-P7 NPs were bactericidal against *Staphylococcus aureus*, *Escherichia coli* and *Pseudomonas aeruginosa*. Selectivity indices values up to 2.4, determined by cytotoxicity experiments on human keratinocytes, suggested that CB1H-P7 NPs could be promising for counteracting serious infections sustained by most of the isolates tested in this study.

## 1. Introduction

MDR pathogens, defined as bacteria resistant to at least three classes of antibiotics, are growing and spreading dramatically worldwide [1,2], thus representing one of the biggest threats to global health and food security [3,4]. They cause infections progressively more difficult to counteract, as the antibiotics become less and less or even no longer effective [5,6,7]. An operative and urgent action is therefore needed to avoid entering a post-antibiotic era, in which even common infections and minor injuries will kill again [8,9], and the achievements of modern medicine, such as organ transplants and complex surgeries, will certainly take place in an unsafe manner. Currently, antimicrobial resistance (AMR) and derived superbugs already take an enormous toll on health-care systems around the world. About 700,000 people globally die each year due to AMR, and worryingly, without new and better treatments, that number could rise to ten million by 2050 [10]. Overcoming antibiotic resistance is an absolute priority for the World Health Organization (WHO) which encourages, through public-private partnerships [9], the research and development of new antibacterial agents capable of acting with mechanisms other than those of antibiotics available, and which have a lower propensity to develop resistance [6].

Unfortunately, in recent years, the growing emergence of antibiotic resistance has been aggravated by a gradual decline in drug discovery [11], due to the lack of incentive for pharmaceutical companies to invest in new projects aimed at discovering new antibiotics [12]. However, the delay in the discovery of new antibiotics is also due to the technical difficulties related to the identification and synthesis of new antibacterial agents. So, a desirable option would be the discovery of new antibacterial model molecules, achievable with low economic inputs. In this scenario, consisting of poor therapeutic options, and technical/economic difficulties, the diazole ring of pyrazole could represent an excellent template molecule for the development of new antibacterial agents, effective where traditional antibiotics fail. Indeed, molecules that contain it have already been shown to be effective on various bacterial strains, including MDR variants [13], and the herbicidal and fungicidal effects by some of these molecules have been reported [14,15,16]. Additionally, ceftolozane, which is a pyrazole-containing semi-synthetic, broad-spectrum, fifth-generation cephalosporin antibiotic is well known to possess bactericidal activity against certain Gram-negative and Gram-positive bacteria. Unfortunately, being inactivated by β-lactamase enzymes produced by several MDR bacteria, it is administered in combination with tazobactam, a β-lactamases inhibitor [17,18]. Regrettably, while tazobactam is capable to prevent ceftolozane inactivation by serine-type β-lactamases, it is unable to protect it from metallo-β-lactamases, thus establishing the inactivity of the ceftolozane/tazobactam combination against bacteria producing, for example, highly effective carbapenemases such as the New Delhi metallo-beta-lactamase (NDM) [17]. In addition to having available a library of pyrazoles, also including previously synthetized molecules, to meet the desired low economic inputs required for the development of new antimicrobial agents and to exploit the promising antibacterial characteristics of pyrazoles, we recently developed a new one-pot, low-cost synthetic strategy for the preparation of highly functionalized pyrazole derivatives [19]. Furthermore, we recently synthetized pyrazole-based dendrimer nanoparticles, developed to address the solubility disadvantages of the pristine CR232 and BBB4 pyrazoles showed in Figure 1.

The obtained water-soluble and clinically applicable BBB4-G4K and CR232-G5K NPs, realized using dendrimers (G4K and G5K) free of any antibacterial effects, demonstrated potent specific or broad-spectrum antibacterial activity also against MDR bacteria, low level of cytotoxicity on eukaryotic cells and high selectivity indices [6,20,21,22]. Interestingly, CR232-G5K NPs were effective also against isolates of *P. aeruginosa* and *K. pneumoniae* resistant to colistin, on which all other clinically approved antibiotics failed and against NDMs, thus being more efficacious than ceftolozane [22]. More recently, aiming at improving the weak antibacterial activity (MICs = 128 µg/mL and >256 µg/mL on selected Gram-positive and Gram-negative isolates respectively) of a new pyrazole hydrochloride salt (CB1H) (Figure 2a), CB1H-loaded nanoparticles (CB1H-P7 NPs) were developed by using a cationic copolymer (P7) (Figure 2b) [23].

In this case, P7 prepared by us [24], had proved to possess very potent broad-spectrum bactericidal activity (MICs = 0.6–4.6 µM and 2.3–9.3 µM against Gram-positive and Gram-negative bacteria, respectively) [25], even if also considerable cytotoxicity on human tumor cells (LD_50_ = 4.1–5.1 µM) [24]. However, given the previously reported selectivity of cationic materials on tumor cells and bacteria and their low cytotoxicity on healthy eukaryotic cells, we did not find these results particularly alarming [26,27,28,29,30,31]. Here, CB1H-P7 NPs have been tested on several clinical isolates of Gram-positive and Gram-negative species, including MDR strains, observing high improvements in the antibacterial properties of both CB1H and P7 by a complex-based cooperation, and strong bactericidal effects already after 1 h of exposure. Particularly, very low MICs were observed for CB1H-P7 NPs, thus establishing that when complexed, P7 was 2-fold more potent that P7 alone against all bacteria tested, while CB1H was 2–16-fold and 4–33-fold more potent than pristine CB1H against Gram-positive and Gram-negative bacteria respectively. Conjecturing a possible clinical use of CB1H-P7 NPs, cytotoxicity experiments on human fibroblasts were performed on P7, CB1H and CB1H-P7 NPs. Even if both P7 and CB1H-P7 NPs proved values of LD_50_ lower than that of pristine CB1H, the selectivity indices (SIs) determined for all bacteria for both compounds were significantly higher than those of CB1H and SIs of CB1H-P7 NPs were higher than those of P7 for the most part of bacteria tested in this study.

## 2. Materials and Methods

### 2.1. Synthesis and Characterization of P7, CB1H and CB1H-P7 NPs

The synthetic procedure to obtain CB1H and the CB1H-loaded P7-based cationic NPs (CB1H-P7 NPs) used in this study, and the details of their characterization, have recently been reported [23], while the synthesis and characterization of P7 was previously reported [24].

### 2.2. Microbiology

#### 2.2.1. Bacterial Species Considered in This Study

Several clinical isolates of different genera of Gram-positive and Gram-negative species for a total of 36 strains were employed in this study. All bacteria belonged to a collection obtained from the School of Medicine and Pharmacy of University of Genoa (Italy). Their identification was carried out as described in our previous works [24,32] Particularly, 24 strains were of Gram-negative species, including isolates of *P. aeruginosa*. *Escherichia coli*, *Enterobacter aerogenes*, *Stenotrophomonas maltophilia*, *Klebsiella pneumoniae* and *Acinetobacter baumannii*, 12 strains were of Gram-positive species, including isolates of the genus *Enterococcus* and of the genus *Staphylococcus*. More information concerning the bacterial species used in this study have been reported in Table 1.

#### 2.2.2. Determination of the Minimal Inhibitory Concentrations (MICs) 

The antibacterial effects on several strains of CB1H and P7 were investigated previously [23,25], by determining their MICs, following the microdilution procedures detailed by the European Committee on Antimicrobial Susceptibility Testing EUCAST [33], as also reported in our previous studies [25,32]. Here, following the same procedure, the antimicrobial activity of CB1H-P7 NPs, CB1H and P7 was investigated by determining their MICs on the same strains, for all samples. for comparison purposes. Particularly, serial 2-fold dilutions of solutions of P7, CB1H, and CB1H-P7 NPs (DMSO), ranging from 1 to 256 µg/mL, were used, while DMSO not containing the tested substances was used as control, to verify the absence of antibacterial activity of the solvent used for the experiments. All MICs were obtained at least in triplicate, and results were expressed reporting the modal value or the value that has been observed most frequently. In case of equivocal or not clear results, more than three determinations of MICs were carried out. Finally, we used the drug loading (DL%) value previously reported for CB1H-P7 NPs [23], to estimate the MICs of P7 and CB1H present in the complex from the MICs obtained for CB1H-P7 NPs. Particularly, since CB1H-P7 NPs contain CB1H at 48.6 % (*w*/*w*) and P7 at 51.4 % (*w*/*w*), the MICs of complexed P7 and CB1H corresponded to the 51.4/100 and to the 48.6/100 of the MICs of NPs, respectively.

#### 2.2.3. Time-Kill Experiments

Killing curve assays for CB1H-P7 NPs were performed on various isolates of *S. aureus, E. coli, K. pneumoniae* and *P. aeruginosa* as previously reported [32,34]. Experiments were performed over 24 h initially at concentrations 4–5 times the MICs, then to compare the potency of CB1H-P7 NPs on the different species, killing curves assays at 4 times the MICs were reperformed on species where 5 × MICs concentrations were initially used.

### 2.3. Evaluation of Cytotoxicity of CB1H, P7 and CB1H-P7 NPs on Human Keratinocites (HaCaT)

#### 2.3.1. Cell Culture

Human keratinocytes (HaCaT), deriving from a generous gift of the Laboratory of Experimental Therapies in Oncology, IRCCS Istituto Giannina Gaslini (Genoa, Italy), were grown as previously described [6,22,35,36].

#### 2.3.2. Viability Assay

The dose-dependent cytotoxicity experiments carried out using P7 on eukaryotic cancer cells (HTLA-230 and HTLA-ER neuroblastoma cells) have been previously reported [24]. Here, HaCaT cells were seeded in 96-well plates (at 4 × 10^3^ cells/well) in complete medium and cultured for 24 h. The seeding medium was removed and replaced with fresh complete medium that had been supplemented with 1, 5, 10, 15, 20, 25, 50, 75 and 100 μM of CB1H, with CB1H-P7 NPs at concentrations providing the concentrations used of CB1H, and with P7 at the concentrations provided by the concentrations used of CB1H-P7 NPs. In detail, the concentrations of each sample administered to cells have been reported in Table 2. Cells were then incubated for an additional 24 h. The effect on cell growth was evaluated by the fluorescence-based proliferation and cytotoxicity assay CyQUANT^®^ Direct Cell Proliferation Assay (Thermo Fisher Scientific, Life Technologies, Monza Brianza, Italy) according to the manufacturer’s instructions and as described in our previous works [6,22,36]. The fluorescence of the samples was measured using the mono-chromator-based M200 plate reader (Tecan, Männedorf, Switzerland) set at 480/535 nm. The experiments were carried out at least three times, and samples were run in quadruplicate.

### 2.4. Statistical Analyses

Statistical significance of differences between experimental and control groups in cytotoxicity studies was determined by a two-way analysis of variance (ANOVA) with the Bonferroni correction. The analyses were performed with Prism 5 software (GraphPad, La Jolla, CA, USA). Asterisks indicate the following *p*-value ranges: * = *p* < 0.05, ** = *p* < 0.01, *** = *p* < 0.001.

## 3. Results and Discussion

The reader interested in knowing the main physicochemical features of CB1H-P7 NPs herein biologically tested can find them in our recent work [23]. Additionally, a table containing this information has been provided in the Appendix A).

### 3.1. Antibacterial Effects of CB1H-P7 NPs

As mentioned, pyrazole derivatives, promising as new antibacterial agents but difficult to evaluate and to apply as antibacterial therapeutics due to their water insolubility, have previously been successfully encapsulated in synthetic cationic dendrimers [20,21]. With this strategy, we obtained pyrazole-loaded water-soluble NPs which were easily evaluated for their antibacterial effects and cytotoxicity on eukaryotic cells [6,22]. Although the dendrimers used as solubilizing agents were free of any antibacterial effect, following the encapsulation, the antibacterial properties of pristine pyrazoles were significantly improved and their cytotoxicity reduced, achieving pyrazole-based NPs endowed with potent antibacterial/bactericidal activity against several species of MDR bacteria [6,22].

Now, we first synthetized a new highly substituted pyrazole derivative, in the form of ammonium hydrochloride salt (CB1H), to resolve the poor water solubility of the free amine in advance [23]. Then, after obtaining considerably high MIC values (MIC = 128 µg/mL and >256 µg/mL on Gram-positive and Gram-negative isolates, respectively), CB1H was formulated into nanoparticles (CB1H-P7 NPs) using a cationic copolymer (P7) with strong broad-spectrum bactericidal effects to enhance its weak antibacterial activity [23]. Here, CB1H-P7 NPs were tested against several isolates of different species of Gram-positive and Gram-negative bacteria, and their MIC values were calculated.

#### 3.1.1. Determination of MIC Values

MIC values were determined on several clinical isolates of different species of Gram-negative and Gram-positive bacteria for a total of 36 isolates, and the results obtained are reported in Table 3 (Gram-positive) and Table 4 (Gram-negative).

To compare the antibacterial effects of compounds with very different molecular weights (MW), we preferred to evaluate their MIC values expressed in µM concentrations rather than in µg/mL, as it is usually done in microbiological studies. However, as in our previous work, we reported MICs in both scales and considered both small molecules such as CB1H and macromolecules, such as CB1H-P7 NPs and P7, as inactive against a specific strain when MIC values > 128 µg/mL were observed, regardless of their very different MW. Consequently, pristine CB1H was weakly active only against the isolates of Gram-positive species (MICs = 128 µg/mL), while it was completely ineffective against all Gram-negative strains tested (MICs ≥ 256 µg/mL) (bold light blue data in the third column of Table 3 and Table 4). On the contrary, CB1H-loaded NPs showed wide antibacterial profiles, thus establishing the success of the nanoencapsulation strategy. Particularly, when tested against 12 MDR isolates of different Gram-positive species, and 24 Gram-negative isolates, CB1H-loaded NPs were inactive against only two very problematic MDR strains of Gram-negative species. In detail, CB1H-P7 NPs were inactive (MIC = 256 µg/mL) on only one out of nine *P. aeruginosa* MDR isolates, which was also resistant to the avibactam-ceftazidime combination, and against one out of four *K. pneumoniae* strains KPCs producers (MIC = 256 µg/mL). In contrast, CB1H-P7 NPs have been shown to be effective against all other isolates of the same genera, including those strains also resistant to colistin (underlined data in Table 4), one of the last remaining antibiotics to treat infections sustained by Gram-negative MDR insensitive to carbapenems. Indeed, NPs displayed very low MICs against a colistin-resistant isolate of *K. pneumonia* (MIC = 2.4 µM) and against a colistin-resistant strain of *P. aeruginosa* (MIC = 1.2 µM). In addition, although empty P7 proved to be a powerful antibacterial agent, CB1H-P7 NPs were two times more potent than P7 on the tested bacteria, regardless of their species or antibiotic resistance. Moreover, while CB1H-P7 NPs were active against the colistin-resistant isolate of *P. aeruginosa* (strain 265), P7 alone was ineffective against it (MIC > 256 μg/mL). The highest antibacterial effects of CB1H-P7 NPs were observed against 11 out of 12 isolates of Gram-positive species (MICs = 0.6–2.4 µM vs. 1.2–4.8 µM of P7) and particularly against *E. faecium* and *S. epidermidis* (MICs = 0.6–1.2 µM vs. 1.2 µM of P7). However, CB1H-P7 NPs proved strong antibacterial effects also against the most part of isolates of Gram-negative species (MICs = 1.2–9.6 µM vs. 1.2–18.6 µM of P7), and especially against all strains of MDR *E. coli* (MIC = 1.2 µM vs. 2.3–4.6 µM of P7), two out of four isolates of KPC-producing *K. pneumoniae* (MIC = 2.4–4.8 µM vs. 4.6–9.3 µM of P7), all *A. baumannii* (MIC = 1.2 µM vs. 2.3–9.3 µM of P7), all *P. aeruginosa* (MIC = 1.2–2.4 µM vs. 1.2–4.6 µM of P7) and *S. maltophilia* (MICs = 1.2–2.4 µM vs. 2.3–4.6 µM of P7). Interestingly, not considering the nano-formulation in its totality but the amount in weight of P7 and CB1H present in the amount of CB1H-P7 NPs which were necessary to inhibit bacteria, we noted that by complexing CB1H with P7, not only had we greatly improved the weak antibacterial effects of pristine CB1H as desired, but unexpectedly, we had also improved those of P7. Indeed, since, as previously reported [23] (Appendix A), CB1H-P7 NPs contain CB1H at 48.6% (*w*/*w*) and P7 at 51.4% (*w*/*w*), the actual MIC values (µg/mL) of CB1H and P7, when in the complex, are those observable in columns 5 and 6 of Table 3 and Table 4. In this regard, the MICs of complexed P7 were lower than those of P7 when administered alone by two times, against 7 out of 12 Gram-positive isolates and on 12 out of 24 Gram-negative, four times against five out of 24 Gram-negative isolates, and 8 times on one of 24 Gram-negative strains (bold red data in columns 2 and 5 of Table 3 and Table 4). As for CB1H, while when it was pristine it was completely inactive on all Gram-negative isolates, except for one KPC-producing *K. pneumoniae* (strain 376), and *P. aeruginosa* 259, once complexed it was active on all other isolates considered (MIC = 15.6–62.2 µg/mL), resulting in being even 2–16.4 times more potent than not complexed CB1H on Gram-positive bacteria (bold light blue data in columns 3 and 6 of Table 3 and Table 4).

The antibacterial potency of CB1H-P7 NPs and P7 and CB1H, when complexed, was compared with that of antibiotics commonly used against Gram-positive (Table 5) and Gram-negative strains tested in this study (Table 6). Since the MWs of CB1H-P7 NPs and P7 are very different from those of commonly used antibiotics, we considered it more correct to compare the MIC values expressed in micromolar concentrations (µM), which provide how much of the test substance equivalents were administered to the bacteria to achieve inhibition. On the contrary, since CB1H, being a small molecule, shares similar MW with antibiotics, we considered it equally correct to make comparisons using the µg/mL scale. Data in bold in columns 2 and 5 in Table 5 and Table 6 established that, at least in vitro, CB1H-P7 NPs developed by us are remarkably more efficient than all the available antibiotics commonly used to treat infections sustained by the bacteria herein considered.

#### 3.1.2. Time-Killing Curves

Time-kill experiments were performed with CB1H-P7 NPs initially at concentrations equal to 4–5 × MICs on different strains of *P. aeruginosa*, *S. aureus*, *K. pneumoniae* (MICs = 4 × MICs) and *E. coli* (MICs = 5 × MICs). The isolates tested included one colistin-resistant isolate of *P. aeruginosa* (strain 265), one *P. aeruginosa* isolated by patients with cystic fibrosis (strain 1V), one pyomelanin-producing *P. aeruginosa* (strain 432), one isolate of *E. coli* producing NDMs (strain 462) and one producing KPCs (strain 477), three MRSA (strains 18, 187 and 195), and three KPCs-producing *K. pneumoniae* (strain 375, 376 and 490). The usage of different MIC concentrations to assess bactericidal activity on some pathogens reflects the idea that, depending on CB1H-P7 NPs pharmacokinetics, pharmacodynamics and toxicity, they may reach very high levels following oral or parenteral administration in urine or if applied locally. The results obtained therefore in time-killing experiments performed at 5 × MICs might better anticipate their potential activity in overcoming infections sustained by the bacterial species taken into consideration. Anyway, to better compare the killing potency of CB1H-P7 NPs on the different bacterial species considered in time killing essays, we carried out kill experiments at 4 × MICs also on the isolates of *E. coli,* observing results identical to those observed at 5 × MICs. 

Figure 3 shows the most representative curves obtained for the strains 1V, 462, 195 and 490 of the species above specified. CB1H-P7 NPs showed a powerful and extremely rapid bactericidal effect against *E. coli* and *K. pneumoniae* isolates, causing a 4-log decrease in the original cell number already after 4 h of exposure to CB1H-P7 NPs. Subsequently, a constant decrease in the number of bacterial cells was observed, until the total elimination of these isolates observed at 24 h. CB1H-P7 NPs showed an even faster and more potent bactericidal effect against *P. aeruginosa* isolate, which was totally extinct after only one hour of exposure and of which no regrowth was observed in the remaining time. A more gradual bactericidal effect, but completed at 24 h, was also observed against *S. aureus*. Overall, the rapid bactericidal profile demonstrated here by the formulation based on P7-CB1H complexation, could certainly deserve further investigation for future clinical use against MDR strains, even insensitive to colistin. 

To date, as far as we know, except for CR232-G5K NPs that we have recently developed [22], no other pyrazole-containing macromolecule has been reported for its bactericidal effects. On the contrary, bactericidal effects have been demonstrated by one hydroxy-coumarin-substituted pyrazole-derived hydrazone derivative against an ATCC strain of MRSA [37]. Concerning clinically approved molecules, bactericidal activity is possessed by ceftolozane, which is a pyrazole-containing recently synthesized broad spectrum fifth-generation semisynthetic cephalosporin. Being quite often inactivated by β-lactamase enzymes produced by several MDR bacteria, ceftolozane is administered in combination with tazobactam, a first generation β-lactamases inhibitor [18]. Unfortunately, while tazobactam is capable of preventing ceftolozane inactivation by serine-β-lactamases, it is not active against NDM-producing bacteria [18]. In this regard, this study describes instead the very powerful and very rapid bactericidal effects of a pyrazole-containing macromolecule, obtained from a pyrazole derivative with weak intrinsic antibacterial effects, on clinically relevant Gram-positive and Gram-negative MDR strains, also including an isolated producer of beta NDM, against which currently no antibiotic/inhibitor combination is clinically approved [18].

### 3.2. Cytotoxicity of P7, CB1H and CB1H-P7 NPs on HaCaT Human Keratinocytes Cells

The selectivity index (SI) measures the capability of a new antibacterial agent to selectively inhibit the bacterial cell without damaging the eukaryotic one. The SI values are given by the ratio between the concentration of antibacterial agent capable to kill the 50% of eukaryotic cells (LD_50_) and the values of MICs for specific bacteria. Thus, in view of a possible cutaneous use of CB1H-P7 NPs, a dose-dependent cytotoxicity study was performed on human keratinocytes (HaCaT) to evaluate the P7 copolymer (used to potentiate the weakly antibacterial activity of CB1H and never tested on eukaryotic normal cells) [24], pristine CB1H and CB1H-P7 NPs. Such experiments were performed on HaCaT keratinocytes mainly because many infections caused by bacteria considered in this study, such as *Staphylococci* are localized to skin and soft tissue [38]. Consequently, to assess the cytotoxicity of CB1H-P7 NPs and their ingredients, we selected human keratinocytes, which are the primary type of cell found in the epidermis, the outermost layer of the skin, and are more susceptible to colonization by bacteria, fungi, and parasites. The cytotoxic activity of pristine CB1H has been studied at concentrations of 1, 5, 10, 15, 20, 25, 50, 75 and 100 µM, while that of CB1H-P7 NPs at the values that provide the used concentrations of CB1H. Additionally, also P7 was studied at the values provided by the concentrations used of CB1H-P7 NPs. Cells were exposed to the samples for 24 h to obtain data compatible with the measured MICs. Cells viability percentages as functions of concentrations of all samples are shown in Figure 4, where on the x axis has been reported the concentration of CB1H used, corresponding also to that of CB1H provided by the amounts of CB1H-P7 NPs administered. 

Results in Figure 4 evidence that no statistical difference existed between the cell viability of the control and that observed upon administration of CB1H-P7 NPs at concentrations providing CB1H up to 25 µM, while statistical difference existed upon administration of pristine CB1H already at a concentration 20 µM. In any case, except for concentration 15 µM, empty P7 administered at the same concentrations as those provided by the quantity administered of CB1H-P7 NP was less cytotoxic than both CB1H-P7 NPs and pristine CB1H. For a concentration of CB1H ≥ 50 µM, CB1H-P7 NPs started to be more cytotoxic than pristine CB1H (cells viability of 46% vs. 51% of CB1H), reducing the cell viability to 8% vs. 20% of CB1H at the max concentration of CB1H 100 µM.

To calculate the SI values (LD_50_/MIC) of pristine CB1H, of CB1H-P7 NPs necessary to provide the same concentrations of CB1H tested, and of P7 when administered at the same concentration provided by the quantity of CB1H-P7 NPs used, we reported the data of cell viability % obtained at 24 h of exposure vs. the concentrations of pristine CB1H, empty P7, and CB1H-P7 NPs used, according to Table 2. The obtained curves have been shown in Figure 5a,c.

Using all point of the curves in Figure 5a,c, the correspondent tendency lines were obtained with the Ordinary Least Squares (OLS) method, which is an optimization technique (or regression) that allows to find a function, represented by an optimal curve (or regression curve), which comes as close as possible to a dataset (typically points of the plane). The linearity of the regression models obtained was assured by the good value of the coefficients of correlations (R^2^), and the equations of the regression models were used to determine the desired LD_50_. Figure 5b,d show the used dispersion graphs, the best fitting linear regression models, and the related equations of all samples (concentrations of CB1H 1–100 µM) used to compute their LD_50_.

The obtained equations, their R^2^ values, the computed values of LD_50_ and the range of SI obtained for all samples calculated according to Equation (1) have been reported in Table 7.
(1)$$\mathrm{SI}\;={\;\mathrm{LD}}_{50}/\mathrm{MIC}$$
where LD_50_ is the lethal dose (µg/mL or µM) of the antibacterial agent against HaCaT cells, and MIC is the minimum inhibiting concentration (µg/mL or µM) displayed for the same molecule against bacteria.

The SI values of CB1H-P7 NPs computed for each isolate used in this study have instead been reported in Table 3 and Table 4.

According to Table 7 and the data of LD_50_, pristine CB1H resulted in the less cytotoxic compound, while P7 and CB1H-P7 NPs possessed comparable cytotoxic effects on HaCaT cells. Anyway, while the SI value of CB1H resulted in being extremely low for all bacteria, due to its extremely weak antibacterial effects, the SI values (SIs) of the CB1H-loaded NPs obtained using P7, except for two strains, were significantly higher. Additionally, the SIs of CB1H-P7 NPs were for 11 out of 12 isolates of Gram-positive species, and for 21 out of 23 isolates of Gram-negative species, higher than those determined for empty P7.

## 4. Conclusions

CB1H-P7 NPs were previously prepared with the aim of enhancing the weak antibacterial effects of CB1H, using P7 as a copolymer matrix, due to its powerful and broad-spectrum bactericidal characteristics. Here, the obtained NPs have been evaluated in vitro for assessing their antibacterial effects, investigating their cytotoxic profile and determining their selectivity indices (SIs). Particularly, CB1H-P7 NPs were assayed on 36 clinical isolated strains, including MDR isolates of different genera of Gram-positive and Gram-negative species, obtaining excellent results. The antibacterial potency ranged from being considerable (MIC = 4.8 µM) to very potent (MICs = 0.6–2.4 µM) against 34 out of 36 Gram-positive and Gram-negative isolates tested.

Furthermore, CB1H-P7 NPs displayed very low MICs against colistin-resistant isolates of *P. aeruginosa* (1.2 µM) and *K. pneumoniae* (2.4 µM), currently untreatable by the available antibiotics. Moreover, CB1H-P7 NPs resulted in being effective against an isolate of *E. coli* producing NDM β-lactamases, which are capable of hydrolyzing all β-lactam antibiotics except for aztreonam and against which no approved inhibitor works.

Determinations of the amounts of P7 and CB1H complexed in the CB1H-loaded formulation, derived from the CB1H-P7 NPs concentrations that inhibited bacteria, provided MIC values for P7 in the range of 8.2–65.8 µg/mL against Gram-positive species and in the range of 16.5–65.8 µg/mL on Gram-negative species. MIC values in the ranges of 7.8–62.2 and of 15.6–62.2 µg/mL against Gram-positive and Gram-negative bacteria, respectively, were determined for CB1H.

In most cases, the MICs of complexed P7 were 2–8-fold lower than those of P7 when administered alone, while the MICs of complexed CB1H were 2–16.4-fold lower than those of pristine CB1H, thus confirming that the nanotechnological strategy that we adopted not only succeeded in greatly improving the weak antibacterial effects of the pyrazole derivative, but also those already considerable of copolymer P7.

Additionally, time-kill experiments carried out on different MDR strains of *P. aeruginosa*, *S. aureus*, *E. coli* and *K. pneuomniae* evidenced that after 24 h of exposure, CB1H-P7 NPs were capable of killing all bacteria tested. The bactericidal effects of CB1H-P7 NPs were very rapid against *E. coli* and *K. pneumoniae*, which were reduced by >4-log after only 4 h of exposure and were even more potent and rapid and against *P. aeruginosa,* killed already after 1 h. From dose-dependent cytotoxicity studies performed for 24 h on human keratinocyte cells (HaCaT) and the determination of the relative SIs of CB1H-P7 NPs, CB1H and P7, it was shown that, by complexing CB1H with P7, not only the antibacterial effects of both the ingredients (CB1H and P7) were improved, but also their SIs. Indeed, CB1H-P7 NPs have SIs up to 3.3-fold higher than those of P7 and up to 6.5-fold higher than those of BC1H. The results obtained in this study establish that the developed CB1H-P7 NPs are a powerful new agent with rapid bactericidal effects on even the most alarming MDR bacteria. Particularly, CB1H-P7 NPs were found to be more promising than pristine pyrazole and even P7 and worthy of future consideration for possible use against difficult-to-treat bacterial species.

## Figures and Tables

**Figure 1 biomedicines-10-01607-f001:**
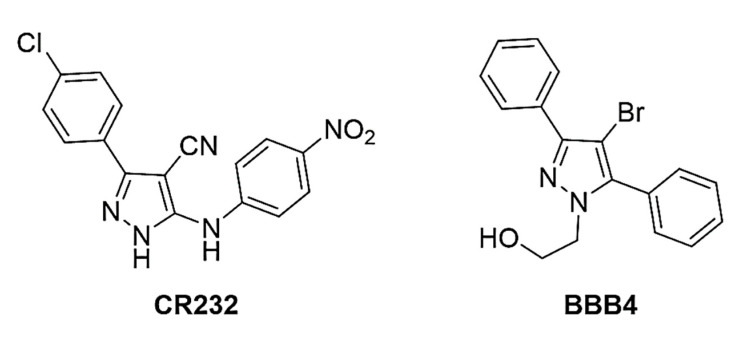
Chemical structure of CR232 and BBB4.

**Figure 2 biomedicines-10-01607-f002:**
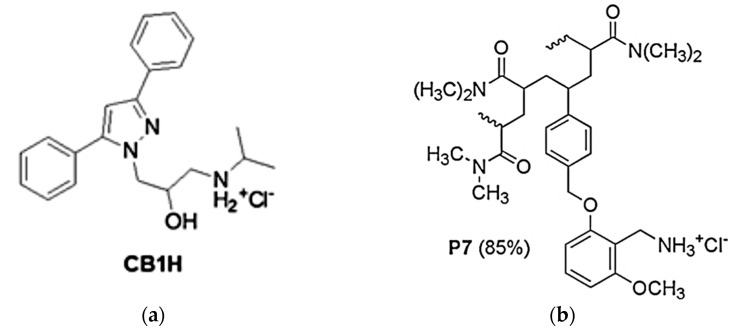
Structure of CB1H (**a**) and of copolymer P7 (**b**).

**Figure 3 biomedicines-10-01607-f003:**
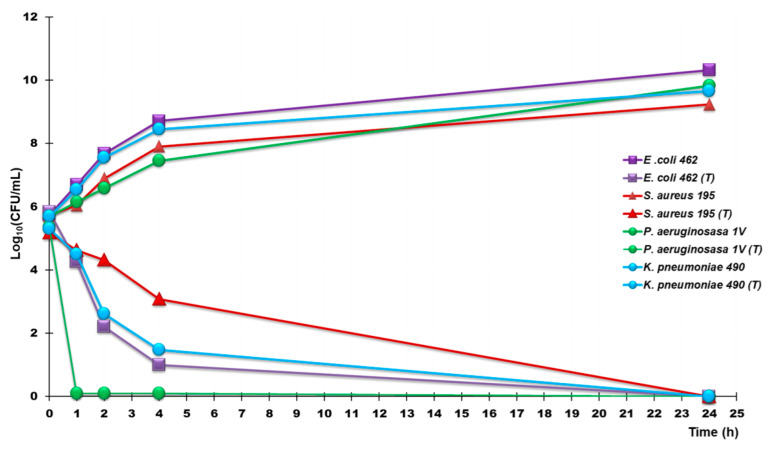
Time-killing curves performed with CB1H-P7 NPs (at concentrations equal to 4 × MIC) on *P. aeruginosa* 1V (green lines), *E. coli* 462 (purple lines), *K. pneumoniae* 490 (light blue lines), and *S. aureus* 195 (red lines). Square indicators denote control samples, while round indicators denote treated samples.

**Figure 4 biomedicines-10-01607-f004:**
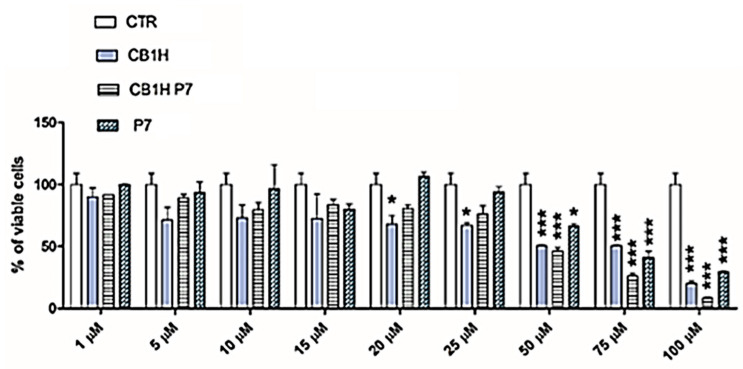
Dose-dependent cytotoxicity activity of P7, CB1H and CB1H-P7 NPs at 24 h towards HaCaT cells. The statistical significance of differences between experimental and control groups was determined via a two-way analysis of variance (ANOVA) with the Bonferroni correction. Asterisks indicate the following *p*-value ranges: * = *p* < 0.05, *** = *p* < 0.001.

**Figure 5 biomedicines-10-01607-f005:**
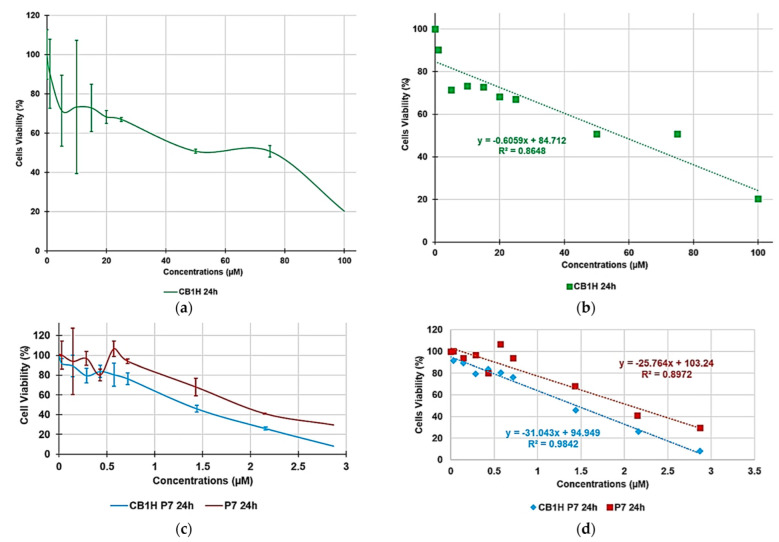
Curves of cell viability (%) vs. concentrations (1–100 µM) of CB1H (**a**) and vs. concentrations of CB1H-P7 NPs providing CB1H 1–100 µM, as well as vs. concentration of P7 as those allowed by the administered CB1H-P7 NPs (**c**) at 24 h towards HaCaT cells; linear regression models fitting the dispersion graphs obtained reporting in graph the cell viability % vs. the concentration of samples at 24 h of exposure of CB1H (**b**), P7 and CB1H-P7 NPs (**d**).

**Table 1 biomedicines-10-01607-t001:** Bacterial species considered in this study.

36 Bacterial Species				
12 Gram-Positive		Resistance Information	24 Gram-Negative	Resistance Information
6 *Enterococci*	3 *E. faecalis*	VRE *	9 MDR *P. aeruginosa* ^#^	1 colistin-resistant, 1 resistant to CAZ-AVI, 3 pyomelanin-producing, 4 from fibrosis cystic patients
	3 *E. faecium*	VRE *		
6 *Staphylococci*	3 *S. aureus*	MRSA	3 *E. coli*	2 KPC-producing, 1 NDM carbapenemase-producing
			1 *E. aerogenes*	Carbapenems-resistant
		MRSE **	4 MDR *S. maltophilia*	2 co-trimoxazole-resistant, 2 co-trimoxazole intermediates
	3 *S. epidermidis*		4 *K. pneumoniae*	KPC-producing *K. pneumoniae* ^§^
			3 MDR *A. baumannii*	1 co-trimoxazole-resistant

VRE = vancomycin resistant; MRSA = methicillin-resistant *S. aureus*; MRSE = methicillin-resistant *S. epidermidis*; CAZ-AVI = ceftazidime-avibactam combination; * 1 was also resistant to teicoplanin; ** two were resistant also to linezolid; ^#^ all resistant to carbapenems; KPCs = β-lactamase enzymes of the KPC family; ^§^ one also resistant to colistin.

**Table 2 biomedicines-10-01607-t002:** Concentrations of each sample administered to HaCaT cells (24 h exposure).

Samples	Concentrations (µM)									
CB1H	0	1	5	10	15	20	25	50	75	100
CB1H-P7	0	0.02874	0.1437	0.2874	0.4311	0.5748	0.7185	1.4370	2.1555	2.8740
P7	0	0.02867	0.1433	0.2867	0.4300	0.5734	0.7167	1.4334	2.1501	2.8668

**Table 3 biomedicines-10-01607-t003:** MIC values of P7 and CB1H when administered alone, MIC values of CB1H-P7 NPs, and those of P7 and CB1H in the complex according to the (DL%) of CB1H-P7 NPs [23] against bacteria of Gram-positive species, obtained from experiments carried out at least in triplicate, expressed as µM and µg/mL.

	Original P7 (13,719) ^1^	Original CB1H (371.9) ^2^	CB1H-P7 NPs (26,624) ^1^	Complexed P7 (13,719) ^1^	Complexed CB1H (371.9) ^2^	
Strains	MIC µM (µg/mL)	MIC µM (µg/mL)	MIC µM (µg/mL)	MIC µM (µg/mL)	MIC µM (µg/mL)	Selectivity Indices ^3^
** *Enterococcus genus* **						
*E. faecalis* 1 *^,#^	** 2.3 (32) **	** 344.2 (128) **	1.2 (32)	** 1.2 (16.4) **	** 41.9 (15.6) **	1.2
*E. faecalis* 365 *	2.3 (32)	** 344.2 (128) **	2.4 (64)	2.4 (32.9)	** 83.6 (31.1) **	0.6
*E. faecalis* 450 *	2.3 (32)	** 344.2 (128) **	2.4 (64)	2.4 (32.9)	** 83.6 (31.1) **	0.6
*E. faecium* 21 *	1.2 (16)	** 344.2 (128) **	1.2 (32)	1.2 (16.4)	** 41.9 (15.6) **	1.2
*E. faecium* 325 *	** 1.2 (16) **	** 344.2 (128) **	0.6 (16)	** 0.6 (8.2) **	** 21.0 (7.8) **	**2.4**
*E. faecium* 341 *^,#^	** 1.2 (16) **	** 344.2 (128) **	0.6 (16)	** 0.6 (8.2) **	** 21.0 (7.8) **	**2.4**
** *Staphylococcus genus* **						
*S. aureus* 18 **	** 4.6 (64) **	** 344.2 (128) **	2.4 (64)	** 2.4 (32.9) **	** 83.6 (31.1) **	0.6
*S. aureus* 187 **	4.6 (64)	** 344.2 (128) **	4.8 (128)	4.8 (65.8)	** 167.2 (62.2) **	0.3
*S. aureus* 195 **	** 4.6 (64) **	** 344.2 (128) **	2.4 (64)	**2.4 (32.9)**	** 83.6 (31.1) **	0.6
*S. epidermidis* 22 **	** 1.2 (16) **	** 344.2 (128) **	0.6 (16)	** 0.6 (8.2) **	** 21.0 (7.8) **	**2.4**
*S. epidermidis* 180 ***	1.2 (16)	** 344.2 (128) **	1.2 (32)	1.2 (16.4)	** 41.9 (15.6) **	1.2
*S. epidermidis* 181 ***	** 1.2 (16) **	** 344.2 (128) **	0.6 (16)	** 0.6 (8.2) **	** 21.0 (7.8) **	**2.4**

^1^ average molecular mass (Mn); ^2^ MW; ^3^ refers to CB1H-P7 NPs; * denotes vancomycin resistant (VRE), *^,#^ VRE resistant also to teicoplanin; ** denotes methicillin resistance; *** denotes resistance toward methicillin and linezolid; red bold data evidence the MICs of original P7 and complexed P7, according to which the latter was 2-fold more potent than the first; light blue bold data evidence the MICs of pristine CB1H and complexed CB1H, according to which the latter was 2- to 16.4-fold more potent than the not complexed one; black bold data in column seven evidence the highest SIs.

**Table 4 biomedicines-10-01607-t004:** MIC values of P7 and CB1H when administered alone, MIC values of CB1H-P7 NPs, and those of P7 and CB1H in the complex according to the DL% of CB1H-P7 NPs [23], against bacteria of Gram-negative species, obtained from experiments carried out at least in triplicate, expressed as µM and µg/mL.

	Original P7 (13,719) ^1^	Original CB1H (371.9) ^2^	CB1H-P7 NPs (26,624) ^1^	Complexed P7 (13,719) ^1^	Complexed CB1H (371.9) ^2^	
Strains	MIC µM (µg/mL)	MIC (µg/mL)	MIC µM (µg/mL)	MIC µM (µg/mL)	MIC µM (µg/mL)	Selectivity Indices ^3^
** *Enterobacteriaceae family* **						
*E. coli* 238 ^#^	** 4.6 (64) **	** >688.4 (>256) **	1.2 (32)	** 1.2 (16.5) **	** 41.9 (15.6) **	1.3
*E. coli* 477 ^#^	** 2.3 (32) **	** >688.4 (>256) **	1.2 (32)	** 1.2 (16.5) **	** 41.9 (15.6) **	1.3
*E. coli* 462 ^§^	** 2.3 (32) **	** >688.4 (>256) **	1.2 (32)	** 1.2 (16.5) **	** 41.9 (15.6) **	1.3
*K. pneumoniae* 375 ^#^	4.6 (64)	** >688.4 (>256) **	4.8 (128)	4.8 (65.8)	** 167.2 (62.2) **	0.3
*K. pneumoniae* 376 ^#^	9.3 (128)	** >688.4 (>256) **	9.6 (256)	9.1 (124.4)	35.4 (131.6)	0.2
*K. pneumoniae* 377 ^#^	** 9.3 (128) **	** >688.4 (>256) **	4.8 (128)	** 4.8 (65.8) **	** 167.2 (62.2) **	0.3
*K. pneumoniae* 490 CR ^#^	** 4.6 (64) **	** >688.4 (>256) **	2.4 (64)	** 2.3 (32.9) **	** 83.6 (31.1) **	0.6
*E. aerogenes* 484 ^##^	** 4.6 (64) **	** >688.4 (>256) **	1.2 (32)	** 1.2 (16.5) **	** 41.9 (15.6)) **	1.3
** *Non-fermenting species* **						
*A. baumannii* 257	** 2.3 (32) **	** >688.4 (>256) **	1.2 (32)	** 1.2 (16.5) **	** 41.9 (15.6) **	1.3
*A. baumannii* 279	** 2.3 (32) **	** >688.4 (>256) **	1.2 (32)	** 1.2 (16.5) **	** 41.9 (15.6) **	1.2
*A. baumannii* 245 ^COR^	** 9.3 (128) **	** >688.4 (>256) **	1.2 (32)	** 1.2 (16.5) **	** 41.9 (15.6) **	1.3
*P. aeruginosa* 1V ^##^	** 2.3 (32) **	** >688.4 (>256) **	1.2 (32)	** 1.2 (16.5) **	** 41.9 (15.6) **	1.3
*P. aeruginosa* 5V ^##^	** 4.6 (64) **	** >688.4 (>256) **	2.4 (64)	** 2.3 (32.9) **	** 83.6 (31.1) **	0.6
*P. aeruginosa* 6V ^##^	** 4.6 (64) **	** >688.4 (>256) **	2.4 (64)	** 2.3 (32.9) **	** 83.6 (31.1) **	0.6
*P. aeruginosa* 7G ^##^	** 4.6 (64) **	** >688.4 (>256) **	2.4 (64)	** 2.3 (32.9) **	** 83.6 (31.1) **	0.6
*P. aeruginosa* 265 CR ^##^	** ≥18.6 (≥256) **	** >688.4 (>256) **	1.2 (32)	** 1.2 (16.5) **	** 41.9 (15.6) **	1.3
*P. aeruginosa* 432 ^py,##^	1.2 (16)	** >688.4 (>256) **	1.2 (32)	1.2 (16.5)	** 41.9 (15.6) **	1.3
*P. aeruginosa* 447 ^py,##^	** 4.6 (64) **	** >688.4 (>256) **	1.2 (32)	** 1.2 (16.5) **	** 41.9 (15.6) **	1.3
*P. aeruginosa* 244 ^py,##^	** 4.6 (64) **	** >688.4 (>256) **	1.2 (32)	** 1.2 (16.5) **	** 41.9 (15.6) **	1.3
*P. aeruginosa* 259 *^,##^	** 9.3 (128) **	** >688.4 (>256) **	9.6 (256)	** 9.1 (124.4) **	35.4 (131.6)	0.2
*S. maltophilia* 2 ^COI^	** 2.3 (32) **	** >688.4 (>256) **	1.2 (32)	** 1.2 (16.5) **	** 41.9 (15.6) **	1.3
*S. maltophilia* 280 ^COI^	** 4.6 (64) **	** >688.4 (>256) **	1.2 (32)	** 1.2 (16.5) **	** 41.9 (15.6) **	1.3
*S. maltophilia* 384 ^COR^	** 2.3 (32) **	** >688.4 (>256) **	1.2 (32)	** 1.2 (16.5) **	** 41.9 (15.6) **	1.3
*S. maltophilia* 390 ^COR^	2.3 (32)	** >688.4 (>256) **	2.4 (64)	2.3 (32.9)	** 83.6 (31.1) **	0.6

^1^ average molecular mass (Mn); ^2^ MW; ^3^ refers to CB1H-P7 NPs; ^#^ denotes KPC-producing isolates; ^§^ denotes New Delhi metallo carbapenemases (NDMs)-producing isolate; *P. aeruginosa, S. maltophilia* and *A. baumannii* were all MDR bacteria; 1V, 5V, 6V, 7G = MDR strains isolated from patients with cystic fibrosis; CR = MDR (*P. aeruginosa*) or KPC-producing (*K. pneumoniae*) strains resistant also to colistin; ^py^ pyomelanin producers; * resistant to the combination avibactam-ceftazidime; ^##^ resistant to carbapenems; ^COR^ co-trimoxazole resistant; ^COI^ co-trimoxazole intermediate; red bold data evidence the MICs of original P7 and complexed P7 according to which the latter was more potent than the first; light blue bold data evidence the MICs of pristine CB1H and complexed CB1H, according to which, while the first one was inactive against all isolates considered (MICs > 128 µg/mL), the latter, except for two isolates, was active against all strains considered; underlined data refer to the MICs of CB1H-P7 NPs active against CR strains of *P. aeruginosa* and *K. pneumoniae*.

**Table 5 biomedicines-10-01607-t005:** MIC values of CB1H-P7 NPs and those of P7 and CB1H in the complex against bacteria of Gram-positive species, obtained from experiments carried out in triplicate, and those of reference antibiotics expressed as µM and µg/mL.

Strains	CB1H-P7 NPs (26,624) ^1^	Complexed P7 (13,719) ^1^	Complexed CB1H (371.9) ^2^	Reference Antibiotics
	MIC µM (µg/mL)	MIC µM (µg/mL)	MIC µM (µg/mL)	MIC µM (µg/mL)
*E. faecalis* 1 *^,#^	**1.2 (32)**	1.2 (16.4)	41.9 (15.6)	**366.3 (128) ^3^**
*E. faecalis* 365 *	**2.4 (64)**	2.4 (32.9)	83.9 (31.1)	**366.3 (128) ^3^**
*E. faecalis* 450 *	**2.4 (64)**	2.4 (32.9)	83.9 (31.1)	**366.3 (128) ^3^**
*E. faecium* 21	**1.2 (32)**	1.2 (16.4)	41.9 (15.6)	**366.3 (128) ^3^**
*E. faecium* 325 *	**0.6 (16)**	0.6 (8.2)	21.0 (7.8)	**366.3 (128) ^3^**
*E. faecium* 341 *^,#^	**0.6 (16)**	0.6 (8.2)	21.0 (7.8)	**366.3 (128) ^3^**
*S. aureus* 18 **	**2.4 (64)**	2.4 (32.9)	83.9 (31.1)	**386.4 (128) ^4^, 1275.5 (512) ^5^**
*S. aureus* 187 **	**4.8 (128)**	4.8 (65.8)	167.2 (62.2)	**386.4 (128) ^4^, 1275.5 (512) ^5^**
*S. aureus* 195 **	**2.4 (64)**	2.4 (32.9)	83.9 (31.1)	**386.4 (128) ^4^, 1275.5 (512) ^5^**
*S. epidermidis* 22 **	**0.6 (16)**	0.6 (8.2)	21.0 (7.8)	**193.2 (64) ^4^, 637.8 (256) ^5^**
*S. epidermidis* 180 ***	**1.2 (32)**	1.2 (16.4)	41.9 (15.6)	**193.2 (64) ^4^, 637.8 (256) ^5^**
*S. epidermidis* 181 ***	**0.6 (16)**	0.6 (8.2)	21.0 (7.8)	**193.2 (64) ^4^, 637.8 (256) ^5^**

^1^ average molecular weight (Mn); ^2^ MW; * denotes vancomycin resistant (VRE); *^,#^ VRE resistant also to teicoplanin; ** denotes methicillin resistant; *** denotes resistance toward methicillin and linezolid; ^3^ ampicillin; ^4^ ciprofloxacin; ^5^ oxacillin; black bold data evidence the MICs that established that, at least in vitro, CB1H-P7 NPs developed by us are remarkably more efficient than all the available antibiotics commonly used to treat infections sustained by bacteria herein considered.

**Table 6 biomedicines-10-01607-t006:** MIC values of CB1H-P7 NPs and those of P7 and CB1H in the complex against bacteria of Gram-negative species, obtained from experiments carried out in triplicate, and those of reference antibiotics expressed as µM and µg/mL.

Strains	CB1H-P7 NPs (26,624) ^1^	Complexed P7 (13,719) ^1^	Complexed CB1H (371.9) ^2^	Reference Antibiotics
	MIC µM (µg/mL)	MIC µM (µg/mL)	MIC µM (µg/mL)	MIC µM (µg/mL)
*E. coli* 238 ^#^	**1.2** (32)	1.2 (16.5)	41.9 (15.6)	**96.6** (32) ^3^
*E. coli* 477 ^#^	**1.2** (32)	1.2 (16.5)	41.9 (15.6)	**96.6** (32) ^3^
*E. coli* 462 ^§^	**1.2** (32)	1.2 (16.5)	41.9 (15.6)	**96.6** (32) ^3^
*K. pneumoniae* 375 ^#^	**4.8** (128)	4.8 (65.8)	167.2 (62.2)	**96.6** (32) ^3^
*K. pneumoniae* 376 ^#^	**9.3** (256)	9.6 (131.6)	334.5 (124.4)	**96.6** (32) ^3^
*K. pneumoniae* 377 ^#^	**4.8** (128)	4.8 (65.8)	167.2 (62.2)	**96.6** (32) ^3^
*K. pneumoniae* 490 CR ^#^	**2.4** (64)	2.3 (32.9)	83.9 (31.1)	**18.5** (16) ^4^
*E. aerogenes* 484 ^##^	**1.2** (32)	1.2 (16.5)	41.9 (15.6)	**96.6** (32) ^3^
*A. baumannii* 257	**1.2** (32)	1.2 (16.5)	41.9 (15.6)	**193.2** (64) ^3^
*A. baumannii* 279	**1.2** (32)	1.2 (16.5)	41.9 (15.6)	**193.2** (64) ^3^
*A. baumannii* 245 ^COR^	**1.2** (32)	1.2 (16.5)	41.9 (15.6)	**193.2** (64) ^3^
*P. aeruginosa* 1V ^##^	**1.2** (32)	1.2 (16.5)	41.9 (15.6)	**76.2** (64) ^5^
*P. aeruginosa* 5V ^##^	**2.4** (64)	2.3 (32.9)	83.9 (31.1)	**76.2** (64) ^5^
*P. aeruginosa* 6V ^##^	**2.4** (64)	2.3 (32.9)	83.9 (31.1)	**76.2** (64) ^5^
*P. aeruginosa* 7G ^##^	**2.4** (64)	2.3 (32.9)	83.9 (31.1)	**76.2** (64) ^5^
*P. aeruginosa* 265 CR ^##^	**1.2** (32)	1.2 (16.5)	41.9 (15.6)	**18.5** (16) ^4^
*P. aeruginosa* 432 ^py,##^	**1.2** (32)	1.2 (16.5)	41.9 (15.6)	**76.2** (64) ^5^
*P. aeruginosa* 447 ^py,##^	**1.2** (32)	1.2 (16.5)	41.9 (15.6)	**76.2** (64) ^5^
*P. aeruginosa* 244 ^py,##^	**1.2** (32)	1.2 (16.5)	41.9 (15.6)	**76.2** (64) ^5^
*P. aeruginosa* 259 *^,##^	**9.6** (256)	9.6 (131.6)	334.5 (124.4)	**76.2** (64) ^5^
*S. maltophilia* 2 ^COI^	**1.2** (32)	1.2 (16.5)	41.9 (15.6)	**117.7** (64) ^6^
*S. maltophilia* 280 ^COI^	**1.2** (32)	1.2 (16.5)	41.9 (15.6)	**117.7** (64) ^6^
*S. maltophilia* 384 ^COR^	**1.2** (32)	1.2 (16.5)	41.9 (15.6)	**117.7** (64) ^6^
*S. maltophilia* 390 ^COR^	**2.4** (64)	2.3 (32.9)	83.9 (31.1)	**117.7** (64) ^6^

^1^ average molecular mass (Mn); ^2^ MW; ^#^ denotes KPC-producing isolates; ^§^ denotes New Delhi metallo carbapenemases (NDMs) producing isolate; *P. aeruginosa, S. maltophilia* and *A. baumannii* were all MDR bacteria; 1V, 5V, 6V, 7G = MDR strain isolated from patient with cystic fibrosis; CR = MDR (*P. aeruginosa*) or KPC-producing (*K. pneumoniae*) strains resistant also to colistin; ^py^ pyomelanin producers; * resistant to the combination avibactam-ceftazidime; ^##^ resistant to carbapenems; ^COR^ co-trimoxazole resistant; ^COI^ co-trimoxazole intermediate ^3^ ciprofloxacin; ^4^ colistin; ^5^ piperacillin tazobactam; ^6^ trimethoprim sulfamethoxazole; black bold data evidence the MICs that established that, at least in vitro, CB1H-P7 NPs developed by us are remarkably more efficient than all the available antibiotics commonly used to treat infections sustained by bacteria herein considered.

**Table 7 biomedicines-10-01607-t007:** Regression equations, R^2^ values, LD_50_ of empty P7, pristine CB1H and CB1H-P7 NPs, as well as the relative SI ranges calculated using Equation (1).

Sample	Equations	R^2^	LD_50_ (µM)	SI	
				Gram-Positive	Gram-Negative
P7	y = −25.7640x + 103.4	0.8972	2.1	0.5–1.8	0.1–1.8
CB1H	y = −0.6059x + 84.712	0.8648	57.3	≤0.4.	≤0.2.
CB1H-P7 NPs	y = −31.0430x + 94.949	0.9842	1.5	0.3–2.4	0.2–1.3

## Data Availability

All data necessary to support reported results are present in the main text of the article herein and in the Appendix A.

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
