# Peer review of "Enhanced Antibacterial Activity of a Cationic Macromolecule by Its Complexation with a Weakly Active Pyrazole Derivative"

_biomedicines, 2022, doi:10.3390/biomedicines10071607_

Round 1

Reviewer 1 Report

General comments: The paper is very well written, with only some minor English and punctuation errors.  The topic is of interest, but the amount of data presented and discussed is still a bit confusing and there are some doubts on what data is from previous publications. I would recommend the authors to revise the paper, so it is fully clear what data is new and to try and ease the analysis of the results for the readers.

Introduction:

- The information on the sentence “However, the technical difficulties associated with the identification of new antibacterial agents also cope with the delay in the discovery of new antibiotics, and the introduction of new antibacterial model molecules, achievable with low economic inputs, represents an effective and desirable option.” Is not clear. Please reformulate.

- Please elucidate the readers to why you bring up ceftolozane by making it clear that this antibiotic possesses a pyrazole ring.

- Line 93: please give the range of MICs or bactericidal concentrations for P7.

- Minor English and punctuation mistakes (e.g.: line 37 “killing” should be “kill”; line 39 missing comma after “surgeries”; line 44 missing comma after “encourages”; line 49 missing something between “resistance” and “has”, probably a comma followed by “the”; line 84 “of” should be “than the”.

Materials and Methods

- I do not think the section 2.1’s title relates with its content. Please reformulate this section to address the chemical synthesis. The cytotoxicity should be addressed in its proper section and the relevance of the chosen cell-line discussed.

-  Consider condensing strain description on section 2.2.1 in a table.

- Section 2.2.2 – could the authors justify the use of the modal value instead of a range value or even confirming the modal with a fourth assay?

- Section 2.4 – concerning the MIC statement, does it mean that all 3 assays were concordant and the results were the same? If so, this is a results statement and should be replaced elsewhere.

- line 113 the colon after “[23]” should be a full stop; line 141 missing “by” between “investigated determining”

Results and Discussion

- Section 3.1 does not depict results of this paper. This could be placed in the introduction and materials’ sections or simply cite the previous work. Hence, Table 2 should not be part of the results of the current manuscript.

- Missing references on the second paragraph of section 3.2.

- Please specify what is DL %.

- In the MIC method, the authors only refer using a DMSO solution of CB1H-P7 NPs, but in tables 3, 4, and 5 they present data for other compounds (including antibiotics). Is this data original or from another previous study? Please clarify this in the manuscript. Also, did the authors perform a DMSA control? Also, please explain how the complexed P7 and CB1H MICs were calculated taking into account the DL in the methodology section.

- Please reformat the header on table 4 so that the compound names are separated.

- Could the authors make the tables more comprehensive by indicating some of the conclusions regarding efficacy that are detailed in the text? For example, bold lettering or other distinction in order to ease the analysis of the results.

- Section 3.2.2 – please justify the use of 4xMIC and 5xMIC depending on the species analysed.

- line 302 – in what basis did the authors support the statement that the bactericidal effect was “extremely rapid”.

- I suggest the authors edit Figure 3 in order to more clearly distinguish between the controls and treatments.

- Discuss the relevance of the chosen cell line. Please elaborate on the possible cutaneous use…

- Figure 4 should not contain title within the image. Also, the legend on the image could be simplified by removing the “24h”

- line 356 – “both than” should be “than both”

- line 388 – where is the NA on the table?

Author Response

Reviewer 1

General comments: The paper is very well written, with only some minor English and punctuation errors. The topic is of interest, but the amount of data presented and discussed is still a bit confusing and there are some doubts on what data is from previous publications. I would recommend the authors to revise the paper, so it is fully clear what data is new and to try and ease the analysis of the results for the readers.

Introduction:

- The information on the sentence “However, the technical difficulties associated with the identification of new antibacterial agents also cope with the delay in the discovery of new antibiotics, and the introduction of new antibacterial model molecules, achievable with low economic inputs, represents an effective and desirable option.” Is not clear. Please reformulate.

We thank the Reviewer for his comment. As requested, the indicated sentence has been reformulated. Please see lines 51-56.

- Please elucidate the readers to why you bring up ceftolozane by making it clear that this antibiotic possesses a pyrazole ring.

We thank the Reviewer for his comment which enabled us to be clearer for Biomedicines readers. Please, find the requested elucidation at line 62.

- Line 93: please give the range of MICs or bactericidal concentrations for P7.

As requested, the ranges of MICs for P7 have been added (lines 97-98).

- Minor English and punctuation mistakes (e.g.: line 37 “killing” should be “kill”; line 39 missing comma after “surgeries”; line 44 missing comma after “encourages”; line 49 missing something between “resistance” and “has”, probably a comma followed by “the”; line 84 “of” should be “than the”.

We thank the Reviewer for having signalled all these errors. The mistakes have been corrected as requested.

Materials and Methods

- I do not think the section 2.1’s title relates with its content. Please reformulate this section to address the chemical synthesis. The cytotoxicity should be addressed in its proper section and the relevance of the chosen cell-line discussed.

The title section, and the section have been reformulated.

-  Consider condensing strain description on section 2.2.1 in a table.

The request of the Reviewer has been satisfied. Please, see the newly added Table 1.

- Section 2.2.2 – could the authors justify the use of the modal value instead of a range value or even confirming the modal with a fourth assay?

According to the common practice in microbiology, MICs determination are usually carried out in triplicate or more and the numerical value that is reported as MIC corresponds to the values which has been obtained more frequently, that is a modal value. Anyway, a specification has been included in the manuscript (lines 160-163), and in the caption of Tables 3 and 4.

- line 113 the colon after “[23]” should be a full stop; line 141 missing “by” between “investigated determining”

The issues signalled have been met.

Section 2.4 – concerning the MIC statement, does it mean that all 3 assays were concordant and the results were the same? If so, this is a results statement and should be replaced elsewhere.

We apologise to the Reviewer, but the assertion was a misprint. It has now been removed (lines 200-201).

Results and Discussion

- Section 3.1 does not depict results of this paper. This could be placed in the introduction and materials’ sections or simply cite the previous work. Hence, Table 2 should not be part of the results of the current manuscript.

As suggested by the Reviewer, section 3.1 and Table 2 have been removed by the results of our manuscript. Particularly, Table 2 has been moved in a Supplementary Materials file and named Table S1.

- Missing references on the second paragraph of section 3.2.

As requested, the due reference has been included in the second paragraph of section 3.2 (lines 222 and 226).

- Please specify what is DL %.

DL% has been specified at its first mention. Please, see line 163.

- In the MIC method, the authors only refer using a DMSO solution of CB1H-P7 NPs, but in tables 3, 4, and 5 they present data for other compounds (including antibiotics). Is this data original or from another previous study? Please clarify this in the manuscript. Also, did the authors perform a DMSA control? Also, please explain how the complexed P7 and CB1H MICs were calculated taking into account the DL in the methodology section.

We apologise to the Reviewer for our incorrectness concerning the description of MIC method in Section 2. Indeed, although some data about the MIC values of P7 and CB1H had been already reported in previous our works, to have comparable data their MICs were determined on the same strains used for CB1H-P7 NPs using for all compounds DMSO. This explanation has been added in Material and Methods section. Additionally, we confirm to have performed a DMSO (not DMSA) control. This necessary information has been added in Section 2.2.2. Please, see at lines 157-160.

- Please reformat the header on table 4 so that the compound names are separated.

The header of Table 4 has been reformatted as requested.

- Could the authors make the tables more comprehensive by indicating some of the conclusions regarding efficacy that are detailed in the text? For example, bold lettering or other distinction in order to ease the analysis of the results.

We thank the Reviewer for his suggestion, which has been applied in all Tables.

- Section 3.2.2 – please justify the use of 4xMIC and 5xMIC depending on the species analysed.

The justification has been included. Please, see lines 331-337.

- line 302 – in what basis did the authors support the statement that the bactericidal effect was “extremely rapid”.

The justification of our assertion is already included in the text. Please, read the following sentence present in the manuscript:

“CB1H-P7 NPs showed a powerful and extremely rapid bactericidal effect against E. coli and K. pneumoniae isolates, causing a 4-log decrease in the original cell number already after 4 hours of exposure to CB1H-P7 NPs”. Four hours to cause a decrease of 4-log in the number of living bacterial cell can be considered a very short time.

- I suggest the authors edit Figure 3 in order to more clearly distinguish between the controls and treatments.

Figure 3 has been edited and the legend has been modified. The distinction between the controls and treated is evident, just because of the rapid and unequivocal bactericidal potency of our NPs. Additionally, we believe that the use of the same colour for the control and treated strain is useful to highlight the different behaviour of the specific bacterium in presence of absence of the antibacterial agent. Moreover, different colours have been used to differentiate the strains.

- Discuss the relevance of the chosen cell line. Please elaborate on the possible cutaneous use…

The request of the Reviewer has been satisfied. Please, see at lines 379-384.

- Figure 4 should not contain title within the image. Also, the legend on the image could be simplified by removing the “24h”

Figure 4 has been modified according to the Reviewer’s requests.

- line 356 – “both than” should be “than both”

The error has been corrected (line 403).

- line 388 – where is the NA on the table?

The footnote has been removed (line 444).

Reviewer 2 Report

I have suggestions for improvement in writing, presentation, and data interpretation. 

The authors present another example in which a pyrazole-based compound has been complexed with another molecule to produce loaded nanoparticles. While intrinsically interesting, there are a number of problems with this manuscript.

MIC values were obtained against a wide panel of antibiotic resistant bacteria using the pyrazole, the complexed molecule, and the nanoparticle without the pyrazole. Comparing the results was difficult because the complexed nanoparticles contained about 35 molecules of pyrazole compound CB1H per each complex, so the authors wisely chose to include results in both µg/mL and µM. In my opinion, the authors’ interpretation of these results is flawed. It is impossible to separate out the contribution of each component of the complex because in most cases (due to hydrophobicity) the pyrazole compound had little efficacy, especially compared to the nanoparticle material P7. Therefore, although one can argue that in some cases the presence of CB1H (at 35x the P7 concentration) improved the antibacterial action of P7 (generally by 2x, which is one dilution different in an MIC test), it is illogical to claim that the complex had any benefit to the action of CB1H given that P7 alone was many times more effective than CB1H.

It also seems irrelevant to go into fine detail over small improvements in the Selectivity Index (SI) when the SI is terribly small; any concentration of these compounds that are effective against bacteria are toxic to cells at only a few times higher concentration. While such poor SI doesn’t necessarily invalidate the work (providing a blueprint for further modifications), most of the discussion of subtle differences in the SI values are not valuable to this manuscript.

Lines 37, 52, 83 : English language problems.

Line 184: This is not an appropriate way to present this information. There is no description of these graphs in the text nor is there an adequate legend. While the EM pictures are interesting, there is no explanation of why uncomplexed P7 is clumped and CB1H-P7 is not. Presumably these data were previously reported. Only those data that are pertinent to interpreting the results of this manuscript should be included, and those need to be explained in more detail. It is not clear why a model describing the release of CB1H from complexes is a double natural log plot instead of a semi-log plot (linear time). If the graph is to be used in this paper, it needs an explanation. Finally, the title of the first graph in this figure indicates the FTIR of complexed nanoparticles from a different nanoparticle. Is this a misprint or is this a graph from a different paper?

Line 191: The manuscript refers to “pyrazoles-loaded water soluble NPs”. If CB1H-P7 is water soluble, what is shown in the EM pictures in “Table 3” (this is a figure, not a table)? Non-soluble particles are clearly shown.

Line 193: What are “the dendrimers used as solubilizing agents”? The manuscript says they “were free of any bacterial effect”. How are these different from P7 which is shown to have considerable antibacterial activity? This is not clear.

Tables 3, 4, 5, and 6 have considerable notations to indicate what these bacterial strains are resistant to. However, since the mode of action of neither CB1H nor P7 is mentioned, these extensive notes are useless information because we can deduce no correlation between the type of resistance and the action of CB1H-P7. The only relevance is that the strains tested happen to be multidrug resistant. This information could be summarized in a paragraph in the Materials and Methods section rather than cluttering up the tables with symbols. Also, in these tables, the authors attempt to estimate the MIC using the concentrations of CB1H and P7 from the complexes. These values are not measured but are calculated and depend entirely on the accurate and consistent determination of the amount of CB1H released from the complexes, and this is not explained to the degree necessary to accept the correctness of these results. CB1H appears to be released over a period of 12 hours (“Table 3”), complicating the interpretation. Finally, see my earlier comment: given the large contribution to the MIC from P7, the contribution of CB1H in these measurements is impossible to determine, regardless of whether theoretical calculations can be carried out.

Line 313: The format for Figure 3 is inappropriate. The symbols should be described in the actual legend, different symbols should be used for different samples, and the symbol explanations should be in some sensible order.

Line 317: The authors do not cite appropriate literature in their field. There may not be other examples of pyrazole compounds complexed to macromolecules that are bactericidal, but there certainly are bactericidal pyrazole derivatives. Since the authors describe one non-macromolecular example (and its shortcomings) perhaps they should mention others,
e.g.
Molecules 2020, 25, 2758;doi:10.3390/molecules25122758

Line 366: Figure 5b. It appears this figure shows a mathematical model for Figure 5a. However, the result is clearly not linear, not crossing the Y axis anywhere near 100%, and there is no particular reason to think that these results would follow a linear relationship. A reasonable looking R^2 value was obtained by incorrectly plotting the averages of the data in Fig. 5a. ALL the data (as shown by the error bars) should have been included to create the correct equation and R^2 value. I suspect the same improper calculation was done in Fig. 5d.

Author Response

Reviewer 2

I have suggestions for improvement in writing, presentation, and data interpretation.

The authors present another example in which a pyrazole-based compound has been complexed with another molecule to produce loaded nanoparticles. While intrinsically interesting, there are a number of problems with this manuscript.

MIC values were obtained against a wide panel of antibiotic resistant bacteria using the pyrazole, the complexed molecule, and the nanoparticle without the pyrazole. Comparing the results was difficult because the complexed nanoparticles contained about 35 molecules of pyrazole compound CB1H per each complex, so the authors wisely chose to include results in both µg/mL and µM. In my opinion, the authors’ interpretation of these results is flawed. It is impossible to separate out the contribution of each component of the complex because in most cases (due to hydrophobicity) the pyrazole compound had little efficacy, especially compared to the nanoparticle material P7. Therefore, although one can argue that in some cases the presence of CB1H (at 35x the P7 concentration) improved the antibacterial action of P7 (generally by 2x, which is one dilution different in an MIC test), it is illogical to claim that the complex had any benefit to the action of CB1H given that P7 alone was many times more effective than CB1H.

We kindly point out to the Reviewer that usually pyrazole compounds are hydrophobic and poorly soluble in water as the Reviewer asserts in his comment, but not in the case of CB1H which, being in the form of hydrochloride salt, is highly hydrophilic and water soluble like P7 which is a cationic copolymer. For the rest, we fully understand the Reviewer's disquisition, but we cannot deny the evidence. We observed that following the complexing operation, an increase in both CB1H and P7 activity was observed.

It also seems irrelevant to go into fine detail over small improvements in the Selectivity Index (SI) when the SI is terribly small; any concentration of these compounds that are effective against bacteria are toxic to cells at only a few times higher concentration. While such poor SI doesn’t necessarily invalidate the work (providing a blueprint for further modifications), most of the discussion of subtle differences in the SI values are not valuable to this manuscript.

Part of the discussion has been removed by the manuscript. Please, see lines 449-452 and 454-459.

Lines 37, 52, 83 : English language problems.

The English language problems have been solved.

Line 184: This is not an appropriate way to present this information. There is no description of these graphs in the text nor is there an adequate legend. While the EM pictures are interesting, there is no explanation of why uncomplexed P7 is clumped and CB1H-P7 is not. Presumably these data were previously reported. Only those data that are pertinent to interpreting the results of this manuscript should be included, and those need to be explained in more detail. It is not clear why a model describing the release of CB1H from complexes is a double natural log plot instead of a semi-log plot (linear time). If the graph is to be used in this paper, it needs an explanation. Finally, the title of the first graph in this figure indicates the FTIR of complexed nanoparticles from a different nanoparticle. Is this a misprint or is this a graph from a different paper?

We apologise with the Reviewer, but we struggled to understand his comments because EM was used for SEM. We think he referred to Table 2 (original manuscript), which in his following comment he called Table 3. Anyway, Table 2 has been removed by the manuscript, on request of another Reviewer. Concerning the last part of the Reviewer’s comments, the misprint has been corrected, but we note that the misprint was not in the title of the first graph but in the title of the Table. We thank the Reviewer for his report.

Line 191: The manuscript refers to “pyrazoles-loaded water soluble NPs”. If CB1H-P7 is water soluble, what is shown in the EM pictures in “Table 3” (this is a figure, not a table)? Non-soluble particles are clearly shown.

We apologize again to the Reviewer, but we struggle to understand his comment and what it refers to, because the Reviewer talks about Table 3 in place of Table 2 and of EM in place of SEM. Additionally, we found bizarre that he retained our nanoparticles water-insoluble from the SEM image. The SEM does not provide information concerning the solubility of a solid. We remember that the scanning electron microscope (SEM) uses a focused beam of high-energy electrons to generate a variety of signals at the surface of solid specimens and provides information concerning the morphology and dimensions of particles. Furthermore, we have done all the experiments such as DLS, zeta potential, release kinetics, titrations in PBS or water, just since CB1H-P7 NPs are soluble in water. Additionally, Table 3 contains MIC values and not SEM images, perhaps the Reviewer intended Table 2.

Line 193: What are “the dendrimers used as solubilizing agents”? The manuscript says they “were free of any bacterial effect”. How are these different from P7 which is shown to have considerable antibacterial activity? This is not clear.

We make kindly note to the Reviewer that the dendrimers used as solubilizing agents and not possessing antibacterial effects are macromolecules used previously (as reported in line 190, PDF original manuscript) to solubilize (upon encapsulation) other pyrazole molecules (Ref. 20,21). Additionally, we explain to the Reviewer that, because they are dendrimers, they are necessarily very different from P7 which is a copolymer. Anyway, we think that it is not necessary to explain in this manuscript the structure of the dendrimers previously used by us but we can certainly recommend to the Reviewer reading our previous articles, cited in the manuscript and reported below, where the structures of the dendrimers are represented.

  1. Alfei, S.; Spallarossa, A.; Lusardi, M.; Zuccari, G. Successful Dendrimer and Liposome-Based Strategies to Solubilize an Antiproliferative Pyrazole Otherwise Not Clinically Applicable. Nanomaterials 2022, 12, 233. DOI: 10.3390/nano12020233.
  2. Alfei, S.; Brullo, C.; Caviglia, D.; Zuccari, G. Preparation and Physicochemical Characterization of Water-Soluble Pyrazole-Based Nanoparticles by Dendrimer Encapsulation of an Insoluble Bioactive Pyrazole Derivative. Nanomaterials 2021, 11, 2662. DOI: 10.3390/nano11102662.

Tables 3, 4, 5, and 6 have considerable notations to indicate what these bacterial strains are resistant to. However, since the mode of action of neither CB1H nor P7 is mentioned, these extensive notes are useless information because we can deduce no correlation between the type of resistance and the action of CB1H-P7. The only relevance is that the strains tested happen to be multidrug resistant. This information could be summarized in a paragraph in the Materials and Methods section rather than cluttering up the tables with symbols.

For a microbiologist the problem of antibiotic resistance is extremely important and is not easily generalizable in the MDR definition. Otherwise, the type of resistance is important, because it gives an idea of the difficulty in treating the infections the resistant bacteria cause. Bacteria develop progressive resistance. Clinically speaking, a bacterium resistant to carbapenem is worse than one resistant to penicillin. We ask the Reviewer not to force us to modify the Tables, simplifying the resistance of the bacteria tested, because they are actually relevant from a microbiological point of view.

Also, in these tables, the authors attempt to estimate the MIC using the concentrations of CB1H and P7 from the complexes. These values are not measured but are calculated and depend entirely on the accurate and consistent determination of the amount of CB1H released from the complexes, and this is not explained to the degree necessary to accept the correctness of these results. CB1H appears to be released over a period of 12 hours (“Table 3”), complicating the interpretation. Finally, see my earlier comment: given the large contribution to the MIC from P7, the contribution of CB1H in these measurements is impossible to determine, regardless of whether theoretical calculations can be carried out.

We make kindly note to the Reviewer that he goes on talking of Table 3 in place of Table 2. Additionally, we explain that our calculation (as explained at lines 163-167) were based on the drug loading (DL%) assessed previously (Ref. 23) for CB1H-P7 NPs and the obtained data are independent from the release of CB1H from NPs. Additionally, why the Reviewer is so sure that the antibacterial effects of CB1H-P7 NPs depends mostly on the presence of P7 and isn't it instead due to the coexistence of the two ingredients? How does the Reviewer explain that P7's activity has also increased? P7 alone worked less than CB1H-P7 NPs. We observed that there was a cooperation between CB1H and P7.

Line 313: The format for Figure 3 is inappropriate. The symbols should be described in the actual legend, different symbols should be used for different samples, and the symbol explanations should be in some sensible order.

Figure 3 has been modified as requested.

Line 317: The authors do not cite appropriate literature in their field. There may not be other examples of pyrazole compounds complexed to macromolecules that are bactericidal, but there certainly are bactericidal pyrazole derivatives. Since the authors describe one non-macromolecular example (and its shortcomings) perhaps they should mention others,

e.g. Molecules 2020, 25, 2758;doi:10.3390/molecules25122758

The Reviewer is right. The suggested citation has been added (lines 357-359).

Line 366: Figure 5b. It appears this figure shows a mathematical model for Figure 5a. However, the result is clearly not linear, not crossing the Y axis anywhere near 100%, and there is no particular reason to think that these results would follow a linear relationship. A reasonable looking R^2 value was obtained by incorrectly plotting the averages of the data in Fig. 5a. ALL the data (as shown by the error bars) should have been included to create the correct equation and R^2 value. I suspect the same improper calculation was done in Fig. 5d.

We kindly point out to the Reviewer that (as he himself suggested) all the data, i.e. all the points (concentration vs. percentage of live cells) used to construct the curves in Figures 5a and 5c were used to obtain the trend lines in Figures 5b and 5d, which can be considered linear, given the values of R2. These trend lines, their equations and R2 values were constructed with the least square method by Microsoft's Excel software which worked on all the data of the corresponding curves in Figures 5a and 5c. Anyway, for more clarity an explanation concerning this question has been included in the revised version of our manuscript (lines 421-435).

Round 2

Reviewer 1 Report

I still thin section 2.1 is not right. The former publications should be cited in the respective section, i.e. synthesis section, antibacterial section, cytotoxicity section, in order to show readers that the methodology used was the same. I will leave 2.1 just for synthesis and put the other info in their proper section.

Although I understand the use of the modal value for MIC assays, I do not consider it correct. Since MIC assays only test two-fold dilutions of the antimicrobials, it is possible that the true MIC value falls within two consecutive dilutions and hence some assays can give one value and others the above or below concentration. I would consider, at least, including the ranges of MIC values for those cases were different results were obtained in the supplementary material and make the analysis within the paper with the model values for simplification.

The formatted evidenced in the tables (colours, underlines, bolds, etc) must be referred in the legend of the table itself and not only on the text.

The justification given in lines 331-337 does not seem sufficient. The authors should explain why a MICx5 was used for E. coli 462, for example, and a MICx4 was used for P. aeruginosa 1V. We are talking about the same compound used in different species. To compare its killing abilities, it seems obvious that the concentrations used would be the same…

Please give literature support to the new sentence in lines 379-381.

In Figure 3, although there was improvement, the authors should not rely in the different outcomes of controls vs treatments to distinguish the two data sets. The authors use both symbols and colours to distinguish the same thing (strain), which is incorrect. You could use colour for that and use the symbols to distinguish between control and treatment, for example.

In Figure 4, is there a % that the authors considered to be a toxicity threshold? I would suggest as such as to this be indicated in the graph by a horizontal line. Also, the quality of the figure seems low…

Author Response

I still thin section 2.1 is not right. The former publications should be cited in the respective section, i.e. synthesis section, antibacterial section, cytotoxicity section, in order to show readers that the methodology used was the same. I will leave 2.1 just for synthesis and put the other info in their proper section.

We apologise with the Reviewer but only now we understand his request. The manuscript has been modified accordingly (lines 112-117, 140-147, and 172-174).

Although I understand the use of the modal value for MIC assays, I do not consider it correct. Since MIC assays only test two-fold dilutions of the antimicrobials, it is possible that the true MIC value falls within two consecutive dilutions and hence some assays can give one value and others the above or below concentration. I would consider, at least, including the ranges of MIC values for those cases were different results were obtained in the supplementary material and make the analysis within the paper with the model values for simplification.

We cannot agree with the Reviewer. Indeed, it is incorrect saying that, due to the serial dilutions required by the method detailed by EUCAST, the “true” MIC value could fall within two consecutive dilutions. As detailed by EUCAST, the MIC value to be provided is the concentration of the first well of a series of wells at increasing concentrations where no growth of bacteria was observed more frequently. In this regard, if the MIC is the concentration of the selected well, it can be a concentration comprised between the serial concentrations of two consecutive wells. We remind to the Reviewer that we are talking about minimum inhibitory concentration, that is the minimum concentration stopping the bacteria growth, and we suggest him to consider more carefully the EUCAST. To give ranges for MICs is incorrect.

The formatted evidenced in the tables (colours, underlines, bolds, etc) must be referred in the legend of the table itself and not only on the text.

We thank the Reviewer for his suggestion. Accordingly, we have inserted in the Tables legends explanations concerning coloured, bold, underlined data in all Tables (lines 229-232, 242-245, 312-314, and 322-324).

The justification given in lines 331-337 does not seem sufficient. The authors should explain why a MICx5 was used for E. coli 462, for example, and a MICx4 was used for P. aeruginosa 1V. We are talking about the same compound used in different species. To compare its killing abilities, it seems obvious that the concentrations used would be the same…

Since after the first revision we supposed the Reviewer could not be satisfied with our explanation, meantime we were waiting for his decision, we have carried out other time-kill experiments at 4 x MIC on E. coli to have results directly comparable to those observed against other species tested. As expected, we found no significant differences. We have modified the main text and the Figure caption, accordingly (lines 162-164, 326-328, and 339-342, and caption of Figure 3.

Please give literature support to the new sentence in lines 379-381.

The requested reference has been included as Ref 38 (line 388).

In Figure 3, although there was improvement, the authors should not rely in the different outcomes of controls vs treatments to distinguish the two data sets. The authors use both symbols and colours to distinguish the same thing (strain), which is incorrect. You could use colour for that and use the symbols to distinguish between control and treatment, for example.

Figure 3 has been modified following the suggestion of the Reviewer. We thank the Reviewer.

In Figure 4, is there a % that the authors considered to be a toxicity threshold? I would suggest as such as to this be indicated in the graph by a horizontal line. Also, the quality of the figure seems low…

We selected the value of DL50 as a possible threshold, and according to the Reviewer suggestion, we have inserted the due horizontal line in the Figure 4.

Reviewer 2 Report

While we might disagree a little on the significance of some of the findings, the work done on this manuscript is highly commendable, and I find my major concerns have been either removed, corrected, or explained. I look forward to seeing this published.

Author Response

While we might disagree a little on the significance of some of the findings, the work done on this manuscript is highly commendable, and I find my major concerns have been either removed, corrected, or explained. I look forward to seeing this published.

We thank the Reviewer for being satisfied with our work of revision and to have supported the publication of our manuscript.